# A Modeled High-Density Fed-Batch Culture Improves Biomass Growth and β-Glucans Accumulation in *Microchloropsis salina*

**DOI:** 10.3390/plants11233229

**Published:** 2022-11-25

**Authors:** Darío Ocaranza, Iván Balic, Tamara Bruna, Ignacio Moreno, Oscar Díaz, Adrián A. Moreno, Nelson Caro

**Affiliations:** 1Centro de Investigación Austral Biotech, Facultad de Ciencias, Universidad Santo Tomás, Santiago 8320000, Chile; 2Centro de Biotecnología Vegetal, Facultad de Ciencias de la Vida, Universidad Andres Bello, Santiago 8370146, Chile; 3Departamento de Acuicultura y Recursos Agroalimentarios, Área Prioritaria de Investigación (API3), Programa Fitogen, Universidad de Los Lagos, Osorno 5311157, Chile; 4Instituto de Ciencias Biomédicas, Facultad de Ciencias de la Salud, Universidad Autónoma de Chile, Santiago 8910060, Chile

**Keywords:** *Microchloropsis salina*, β-glucans, microalgae, fed-batch, biomass

## Abstract

Algae and microalgae are used as a source of different biomolecules, such as lipids and carbohydrates. Among carbohydrates, polysaccharides, such as β-glucans, are important for their application as antioxidants, antisepsis, and immunomodulators. In the present work, the β-glucans production potential of *Microchloropsis salina* was assessed using two different culture conditions: a high-density batch and a modeled high-density fed-batch. From the biochemical parameters determined from these two cultures conditions, it was possible to establish that the modeled high-density fed-batch culture improves the biomass growth. It was possible to obtain a biomass productivity equal to 8.00 × 10^−2^ ± 2.00 × 10^−3^ g/(L × day), while the batch condition reached 5.13 × 10^−2^ ± 4.00 × 10^−4^ g/(L × day). The same phenomenon was observed when analyzing the β-glucans accumulation, reaching volumetric productivity equal to 5.96 × 10^−3^ ± 2.00 × 10^−4^ g of product/(L × day) against the 4.10 × 10^−3^ ± 2.00 × 10^−4^ g of product/(L × day) obtained in batch conditions. These data establish a baseline condition to optimize and significantly increase β-glucan productivity, as well as biomass, adding a new and productive source of this polymer, and integrating its use in potential applications in the human and animal nutraceutical industry.

## 1. Introduction

β-glucans are polysaccharides constituted by D-glucose monomers, joined by type β glycosidic bonds in positions β-(1,2); β-(1,3); β-(1,4); and β-(1,6) [1,2]. They are found as components in plants, fungi, yeasts, and bacteria, among other microorganisms, playing a structural role or as an energy reservoir [1]. These molecules are known to have several interesting properties, such as anti-cancer activity [3], antiseptic effects [4], antioxidant properties [5], decreasing the risk of cardiovascular disease [6], and improving immunological responses in several animals [7,8,9]. This last observation has been reported in studies carried out on salmon and trout species, whereby supplementing fish diets with β-glucans improve its resistance to pathogens [10,11]

Although several microorganisms have been described as producers of β-glucans, algae and microalgae have emerged as potential complementary producers of these molecules [12]. Among this group, the microalgae *Microchloropsis salina* (Ochrophyta, Eustigmatophyceae, Monodopsidaceae) has been presented as the most promising one, due to its capacity to accumulate large amounts of biomass in an autotrophic way, its easy cultivation, and its rapid growth. All these features also make this microorganism an excellent model to produce food supplements for humans or animals [13]. On the other hand, several authors report that both *Nannochloropsis* and *Microchloropsis* species are involved in the synthesis of β-(1,3)-glucans [12,14,15,16,17]. It is also suggested that these polysaccharides could be stored in the form of laminaria or chrysolaminarin [18].

Previous studies showed that *Microchloropsis salina* can accumulate more biomass, lipids, and β-glucans in a nitrogen-rich media than in a nitrogen-limited media [19,20,21,22]. In addition, the best growth rates and production of lipids, and β-glucans are achieved with the addition of a source of inorganic carbon, such as carbon dioxide at 2.0% (*v*/*v*) [21,23,24,25]. In the case of lipid production, it has been shown that, in high-density batch cultures, the growth of the microorganism is inhibited due to high concentrations of substrate (nitrate and phosphate) [26]. To overcome this situation, researchers have proposed to generate two stages of cultures, a stage rich in substrate to obtain an accelerated growth, and a stage of substrate limitation to promote the synthesis of lipids [27]. In addition to this, few studies have aimed to characterize the production of β-glucans in *Microchloropsis* sp. and none of these improve its production parameters, either at a volumetric or specific level [12]. Despite the benefits that β-glucans have and could bring to the industry, *M. salina* has been used mainly to produce lipids for biofuel elaboration [21]. In summary, the novelty of the strategy of using a two-stage fed-batch culture lies in producing large amounts of carbohydrates. In this work, we designed a fed-batch culture system with two stages under nitrogen sufficient and deficient conditions, to evaluate and improve the productivity of β-glucans in *Microchloropsis salina* cultures.

## 2. Results

### 2.1. Kinetic Parameters in an Optimal Environment

To obtain the necessary parameters for modeling the final culture system, low-density batch cultures were carried out, to replicate these results in high-density cultures. As regards the accumulation of biomass and nitrogen consumption, it was observed that the maximum accumulation of biomass occurred on the eighth day of the experiment, reaching 0.62 ± 0.02 g/L (Figure 1). This phenomenon occurred 4 days after the total exhaustion of nitrogen in the culture medium. Based on this result, it was possible to calculate the specific growth rate of the process, given by the relation between the final biomass and the initial conditions of biomass inoculation. The specific growth rate reached values of 7.02 × 10^−3^ ± 1.17 × 10^−4^ h^−1^. In the same way, it was possible to calculate the substrate yield under optimal culture conditions, reaching 6.2 g of biomass/g of nitrate.

In addition, it was possible to calculate the maximum concentration of substrate that *M. salina* was capable of consuming under high-density conditions. Here, it was observed that the maximum biomass accumulation was 2.04 ± 0.04 g/L in dry biomass (Figure 2).

The observed values for growth rate and substrate yield were approximately three times higher than the obtained when growing *M. salina* in a media containing conventional nitrate and phosphate concentrations (75 mg/L and 4.41 mg/L, respectively) but with a much lower specific growth rate of the process (4.99 × 10^−3^ ± 3.70 × 10^−5^ h^−1^), as well as nitrate yield (2.63 g biomass/g nitrate). At the end of the experiment, which was set at 24 days of continuous nitrate feeding, the provided culture reaches a final nitrate concentration of 827 ± 7 mg/L, in which the residual concentration equals 83 ± 3 mg/L. These results indicate that *M. salina,* in the defined conditions, was able to consume 744 ± 4 mg of nitrate in one liter of culture.

### 2.2. Fed-Batch Culture Modeling

Once the desired kinetic parameters were obtained, it was possible to develop a predictive model of a fed-batch system culture. After performing the calculations described below, it was possible to observe that the experimental time for this culture corresponds to approximately 631.9 h, which takes up to 26 days. Using this timing, it was possible to determine the feeding flow for the fed-batch culture which corresponded to 2.4 mL/h to feed 1.5 L of culture medium to the fermenter. By the same analysis it was possible to predict that, at the end of the experimental development, it would have a final biomass equal to 2.8 g/L approximately. The calculation of the culture time (t) in fed batch cultures was made by mathematical modeling using the initial volume of culture (Vo); final volume of culture (V); initial biomass of culture (Xo); initial concentration of substrate as nitrate (Si); final concentration of substrate as nitrate (S); biomass yield related to nitrate (Yx/s); and specific growth rate of the process (μ) integrated into the following equation:V/Vo = 1 − bxo + bxo × exp(μ × t)(1)
where b corresponds to:b = 1/ (Yxs × (si − s))(2)

The parameters used are shown in Table 1.

### 2.3. High-Density Batch vs. High-Density Fed-Batch

#### 2.3.1. High-Density Batch

After performing an analysis of the growth kinetics of *M. salina* under high-density conditions, it was observed that the culture times were longer than those observed in previous experiments with reduced amounts of nitrates. In this case, the cultures had a total duration of 40 days, with a total depletion of nitrogen supplemented at 36 days of culture.

Similar to the results obtained with the pulse experiment, 2.13 ± 0.05 g/L of dry biomass were achieved with these culture conditions (Figure 3). It was observed that the culture did not stop growing on the day of nitrogen depletion but continued accumulating biomass until 4 days after the nitrogen nutrient had been completely depleted.

The specific growth rate of the process decreased compared to low density experiments, reaching values equal to 3.36 × 10^−3^ ± 4.19 × 10^−4^ h^−1^.

The lower specific growth rate obtained by increasing the amount of nitrate and phosphate in the medium suggests that the kinetic growth of *M. salina* was inhibited under these culture conditions. This phenomenon was observed in the biomass productivity for the culture, which reached 5.13 × 10^−02^ ± 4.00 × 10^−3^ g × L^−1^ × day^−1^ and, likewise, in the yield taken from the substrate, which reached 2.88 g biomass/g nitrate; less than the 6.1 g biomass/g nitrate observed in the supplemented cultures with nitrogen.

A biochemical profile of the *M. salina* biomass was generated from the different cultures, measuring β-glucans, lipids, total carbohydrates, and protein levels at 4 days of the exhaustion of nitrogen. In high-density batch conditions, the biomass obtained from the *M. salina* culture accumulated 43.72 ± 1.33 g of lipids per 100 g of dry biomass, this being the major component of the microalgal biomass in this experimental period. The protein content reached up to 10.23 ± 0.68% of the total biomass (Table 2).

The carbohydrate production reached 17.23 ± 0.15% of in relation to total dry microalgal biomass. The production of β-glucans at intracellular level by *M. salina,* was of 8.04 ± 0.14%. (Table 2).

When calculating the volumetric productivity for β-glucans at 4 days of nitrogen depletion in the culture medium, it was established that the microalgae generate 4.10 × 10^−3^ ± 2.00 × 10^−4^ g of product × L^−1^ × day^−1^.

#### 2.3.2. High-Density Fed-Batch

In the fed-batch culture was observed that after 26 days of feeding at constant flow, followed by 4 days of deprivation of nitrogen sources, reached 2.32 ± 0.07 g/L of dry biomass (Figure 4); a value which was below the theoretical definition (2.8 g/L calculated with Equations (1) and (2)). The culture continued accumulating biomass once the nitrogen supplemented in the culture medium was exhausted and a linear trend was observed in the graph (Figure 4).

The culture was maintained with nitrate consumption at limit levels throughout the experimental phase, with the amount of residual nitrate in the culture medium being kept to a minimum level for the microalgae.

The value reached by the nitrate yield in fed-batch mode was higher than in batch cultures grown with the same amount of supplemented nitrate, increasing from 2.88 g of biomass/g of substrate in batch mode to 3.10 g of biomass/g of substrate in a fed-batch.

When analyzing the kinetic parameters obtained, the fed-batch culture recorded a biomass productivity equal to 8.00 × 10^−2^ ± 2.00 × 10^−3^ g/(L × day) and a specific growth rate of the process equals to 4.63 × 10^−3^ ± 5.10 × 10^−5^ h^−1^.

Regarding the biochemical profile of the microalgal biomass obtained (Table 2), it was observed that the microalgal accumulated 40.20 ± 2.20% of lipids. As for carbohydrates and proteins, they accumulated 16.23 ± 1.78% and 12.53 ± 0.92%, respectively. Finally, regarding the accumulation of β-glucans, *M. salina* in high-density fed-batch conditions at 30 days accumulated 7.71 ± 0.43% of the polymer.

#### 2.3.3. Comparative Analysis of Kinetic Parameters

When comparing the cultures performed for the two modalities evaluated (high-density batch (HB) and high-density fed-batch culture (HFB)), it was observed that the kinetic parameters (X, µmax, Qx, Yx/s, and Qp) were different between both modalities.

When analyzing the kinetics of biomass production, it was observed that the biomass values in both culture modes remained at similar levels (Figure 5). In the case of HFB it reached the maximum concentration in a period of 30 days, while the HB reached its peak accumulation in 40 days, showing a growth curve displaced to the right in comparison to the fed-batch culture curve (Figure 5). These results propose that this mode significantly increases the volumetric productivity of biomass and products, such as lipids and β-glucans.

At the level of the specific growth rate of the different processes, it was observed that cultures in high-density batch obtained the lowest values (3.36 × 10^−3^ ± 4.19 × 10^−5^ h^−1^), while fed-batch cultures obtained higher values reflected in higher productivity of biomass of microalgae (4.63 × 10^−3^ ± 5.10 × 10^−3^ h^−1^) (Figure 6A).

As for productivity in biomass, HB again recorded the lowest values, reaching 5.33 × 10^−2^ ± 1.00 × 10^−4^ g/(L × day) (Figure 6B). The values for HFB reached 8.00 × 10^−2^ ± 2.00 × 10^−3^ g/(L × day) (Figure 6B).

As for the yield of the nitrate substrate, there were no significant differences when contrasting this parameter in HFB and HB (3.10 and 2.88 g of biomass/g of nitrate, respectively) (Figure 6C).

Finally, when comparing the volumetric productivity for β-glucans, it was observed that HFB mode reached 6.09 × 10^−3^ ± 7.05 × 10^−6^ g of product/(L × day). This value is significantly higher than that recorded in HB (4.29 × 10^−3^ ± 1.62 × 10^−4^ g of product/(L × day)) (Figure 6D).

## 3. Discussion

The results obtained for the maximum biomass accumulation of *M. salina*, were consistent with the values reported in the literature. In the work carried out by Araujo and collaborators (2019) [28], the biomass production of the microalgae *M. salina* reached a maximum of 2.2 g/L under similar conditions. However, to reach these levels of biomass the time must be increased considerably. This phenomenon is mainly explained by inhibitory effects by the substrates, both nitrates and phosphates, as previously reported [26,29,30].

Furthermore, it is curious to note that the culture stops growing only 4 days after the nitrogen has been completely depleted in the culture medium. Similar situations have been described and shown towards the accumulation of intracellular lipids in stationary growth stages [31,32]

When observing the biochemical parameters, the values for lipid accumulation coincide with those reported in previous studies using this microalgae specie, where the biomass could accumulate up to 50% of lipids in nitrogen-limited conditions [33,34].

Some authors have suggested that this type of performance in *M. salina* is mainly related to the reduction in protein production under metabolic stress. The lack of nutrients, such as nitrates, could affect the growth and development of the microorganism, favoring the production of reserve metabolites [35].

With regard to carbohydrate production, these values are consistent with those previously reported, where the microalgae could accumulate around 20% of carbohydrates [35], being its second major component. In the case of euglenoids, which are capable of accumulating high quantities of paramylon [36], or in *Skeletonema costatum,* which accumulate β-glucans as an energy reservoir [37], it has been suggested that there is a mechanism that diminishes its metabolisms and focuses in accumulate reserve metabolites to maintain cell viability. The full metabolic activities return once the nutrients are restored to requirement levels by the microalgae.

As regards the accumulation of β-glucans, although the data obtained show a lower accumulation than the percentage of yields observed in *Nannochloropsis* sp., which can accumulate up to about 20% of β-glucans [12], our improvement was generated only by modulating a single parameter, such as the culture condition. This is an open opportunity to explore the modification of other parameters, such as light and osmotic stress, to obtain a better production of the metabolite [38].

These improvements have been previously reported in other studies [39], where the influence of the cell density of the inoculum and the concentration of carbon dioxide in fed-batch cultures with *N. oculata* was analyzed. Under these conditions, the researchers observed that the biomass yield for this microalga, which belongs to the same family of *M. salina* [40], was 1.25 times higher in fed-batch conditions than in batch cultures. Additionally, they could maintain the culture conditions and eliminate inhibitory effects due to the high concentration of substrates in the culture medium. If we contrast these results with ours, we concluded that the accumulation of biomass in fed-batch cultures was not significantly higher than in batch cultures under the same growing conditions and substrate concentration, but our volumetric productivity for biomass had an increase close to 1.6 times. A trend is observed when analyzing the specific maximum growth rate, which, in our case, is 3 times higher in fed-batch cultures than in batch cultures. These data confirm the improvement of kinetic parameters when performing fed-batch cultures in comparison to batch cultures. In batch cultures, the inhibitory effects for the development of microorganisms are accentuated, either by a high concentration of substrates or by the production of metabolites that can limit the growth and development of the microorganisms [41]. This phenomenon has also been observed in other types of microalgae, such as *Chlorella zofingiensis*, where fed-batch cultures are used to improve specific growth rates and biomass yield by overcoming inhibitory effects [42]. For example, in the study of Xu et al. (2004) [21], they increased the dry biomass production of *Nannochloropsis* sp. by 40% by adding glucose to the culture medium, reaching 1.1 g/L of biomass over 10 days in fed-batch cultures versus 0.8 g/L in batch cultures. Additionally, under these same conditions, the authors could determine that the amount of lipids was increased in this condition, from 27% to 31% of dry biomass of microalgae.

## 4. Materials and Methods

### 4.1. Chemicals and Reagents

An enzymatic yeast β-glucan kit was obtained from Megazyme and the other chemicals from Sigma-Aldrich (St. Louis, MO, USA). All chemicals and reagents were of analytical grade.

### 4.2. Microalgal Strain and Seed Culture

The strain used was previously obtained from the library of algae CSIRO (Australian Scientific and Industrial Research Organization), Australia. The seed cultures were kept in a 2 L flask (1 L working volume) at 24 °C temperature, 50 µmol m^−2^ s^−1^ of constant light intensity (led DC12V, white light), pH 8.0, and pre-adapted with CO_2_ 2% with an aeration rate equal to 0.2 vvm. These were grown in sterilized artificial seawater (ASW; 29 gr/L NaCl, 1.1 g/L KCl, 0.25 g/L NaHCO_3_, 1.8 g/L CaCl_2_ × 2H_2_O) supplemented with F/2 [43].

### 4.3. Experimental Methodology

To calculate kinetic parameters, such as specific maximum growth rate and substrate yield (nitrogen), *M. salina* was cultured in ASW-F/2 (75 mg/L NaNO_3_) until 8 days after total nitrogen depletion in the culture medium. The amount of accumulated biomass and consumed nitrates were monitored daily. To obtain the maximum nitrogen consumption by the microalgae, *M. salina* was grown in ASW-F/2 modified, adding, every 3 days, pulses of NaNO_3_ (75 mg/L) and NaH_2_PO_4_ (4.41 mg/L) in the amounts used by F/2, until there was a cessation of nitrate consumption by the microalgae. Accumulated biomass and dissolved nitrates were recorded every 3 days. With these data, a system in the fed-batch mode was modeled as follows: *M. salina* was grown in ASW with F/2, modifying the concentration of nitrates (0.744 g/L total; 0.4 g/L of initial substrate; 0.704 g/L in feeding) and phosphates in the same proportion. Once the feeding of the substrate-limited culture was finished at 26 days, the culture was maintained without feeding for 4 days (Appendix A) before starting with the stage of total nitrogen limitation, and biochemical parameters (lipids, proteins, and carbohydrates) were measured at the end of the development stage of the system. The data were contrasted with batch mode crops under the same growing conditions (0.744 g/L NaNO_3_).

All cultures were made in 2 L flasks (2 L working volume) at 24 °C temperature, 50 µmol m^−2^ s^−1^ of constant light intensity (led DC12V, white light), pH 8.0, and with CO_2_ 2% with an aeration rate equal to 0.2 vvm. All the experimental sets were kept in triplicate.

### 4.4. Analytical Methods

#### 4.4.1. Microalgal Growth and Biomass Measurements

Dry weight biomass was determined gravimetrically after drying the microalgal biomass at 60 °C overnight [27]. Nitrates were measured by spectrophotometric determination at 220 and 275 nm wavelengths [44].

#### 4.4.2. β-Glucans Measurements

The Enzymatic Yeast kit β-glucan (Megazyme™, Bray, Ireland) designed to quantify β-glucan in yeast samples was used, which was previously adapted and optimized to work on microalgae in the facilities of Austral Biotech Research Center (Universidad Santo Tomás, Santiago, Chile) combined with the microalgae cell breakage protocol described by J R Cook (1967) [45].

#### 4.4.3. Lipid Measurements

Lipid determination was performed following the protocol for lipid quantification described by Bligh and Dyer (1959) [46] with some modifications. In brief, 50 mg of lyophilized microalgal biomass were added to 2 mL screw-capped microcentrifuge tubes. Then, 1.5 mL chloroform/methanol solution (2:1) was added to perform phase separation, homogenizing in Bead Beater (Bertin Precellys 24) for 1 min and then vortexing for 5 min. Later, the solution was incubated at 40 °C for 10 min and a volume of 0.5 mL of pure chloroform was added, homogenizing again for 1 min, and vortexing for 5 min. It was then incubated again at 40 °C for 10 min and centrifuged at 16,000× *g* for 5 min at 4 °C. Immediately, the supernatant was rescued and placed in 15 mL tubes. The protocol was applied again to the pellet to maximize lipid recovery, except that a volume of 1 mL of H_2_O was added in the final stage, and the pellet was homogenized for 1 min. Afterward, all the contents were transferred to a 15 mL tube, vortexed for 5 min, and centrifuged at 3000× *g* for 15 min. Then, the organic phase was recovered, dried, and weighed. The result was finally expressed in a percentage relation compared to the amount of microalgal biomass initially weighed.

#### 4.4.4. Protein Measurements

To determine the amount of proteins, presented in the microalgae cultures, protein extracts were made according to the procedure proposed by Slocombe et al. (2013) [47] quantifying the proteins using the Bradford reagent according to the instructions of the kit (Bio-Rad Protein Assay, Hercules, CA, USA).

#### 4.4.5. Total Sugar Measurements

The dry biomass samples were prepared using the protocol described by Richmond and Hu (2013) [48] where the cells were broken, and pigments were removed as suggested by this protocol. The subsequent hydrolysis and quantification of sugars were performed using the protocol described by Yemm and Willis (1954) [49] using the anthrone reagent to perform the measurements.

## 5. Conclusions

In this research, it was possible to characterize the growth of *Microchloropsis salina* in low- and high-density batch cultures, which allows establishing parameters for the modeling of a two-stages fed-batch culture. Together with the modification of environmental parameters, it has proved to be an effective alternative to increase the productivity of metabolites of interest such as lipids, carbohydrates, and β-glucan. It also obtains high yields of biomass, shortening the periods of experimentation without having to modify genetics to meet these objectives. In this work, it was possible to establish baseline conditions to optimize β-glucan productivity, significantly increasing its productive parameters, as well as those of microalgae growth and other biochemical parameters, including lipids, such as EPA, and total carbohydrates.

## Figures and Tables

**Figure 1 plants-11-03229-f001:**
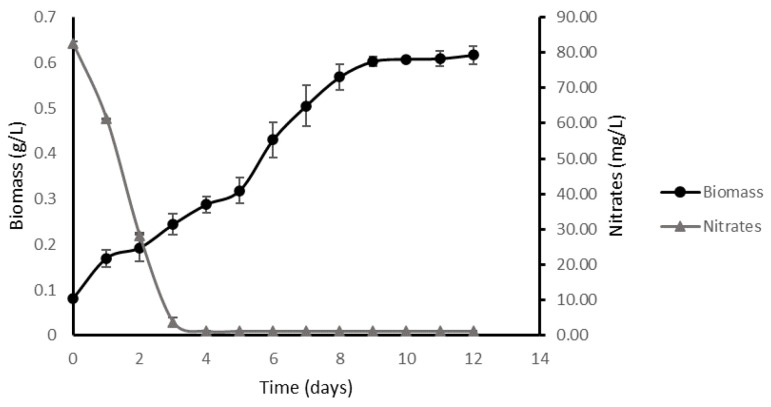
Growth kinetics of *M. salina* in low-density batch cultures. Growth curve expressed in dry microalgae biomass (g/L) and dissolved nitrates (mg/L) in batch culture under low-density conditions as a function of time. The culture was harvested 8 days after the total nitrogen depletion in the culture medium.

**Figure 2 plants-11-03229-f002:**
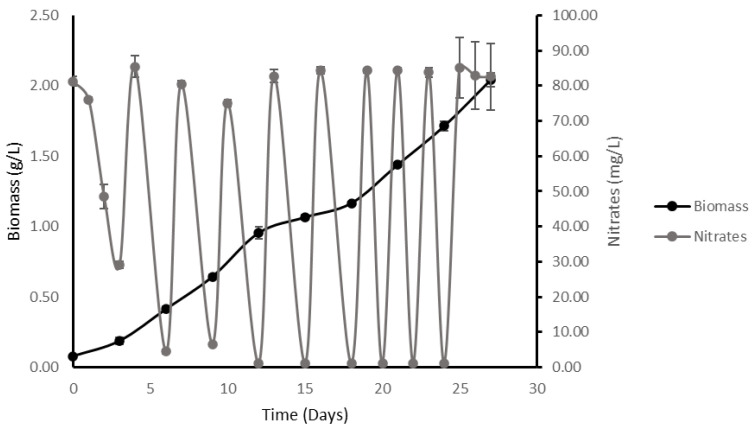
Growth kinetics of *M. salina* with excess nitrates. Culture supplemented by nitrate and phosphate pulses at 75 mg/L and 4.41 mg/L, respectively. The growth curve is expressed in dry microalgae biomass (g/L) and dissolved nitrates (mg/L) in culture as a function of time.

**Figure 3 plants-11-03229-f003:**
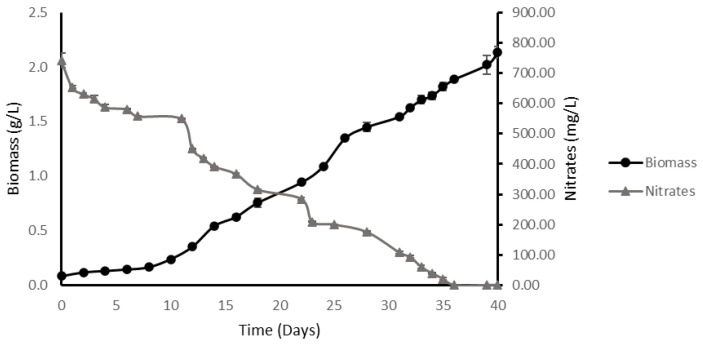
Growth kinetics of *M. salina* in high-density cultures. These cultures were made with a concentration of nitrates provided in the culture medium equal to 744 mg/L and 43.7 mg of initial phosphates. Growth curve is expressed in dry biomass (g/L) and dissolved nitrates (mg/L) as a function of time.

**Figure 4 plants-11-03229-f004:**
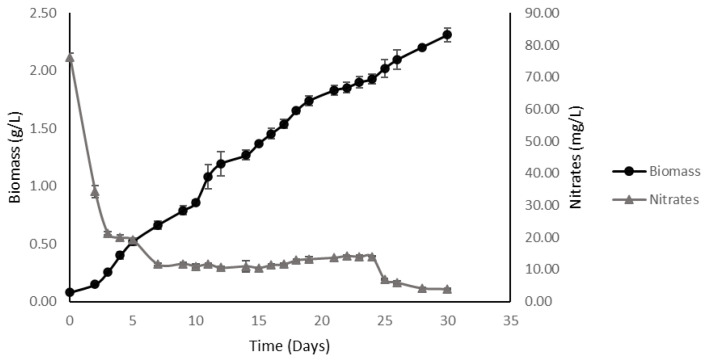
Growth kinetics of *M. salina* in fed-batch cultures. Cultures were fed with the same culture medium generated for batch cultures under high-density conditions, feeding them with a flow rate equal to 1.5 mL/h. Growth curve expressed in microalgal dry biomass (g/L) and dissolved nitrates (mg/L) in culture as a function of time.

**Figure 5 plants-11-03229-f005:**
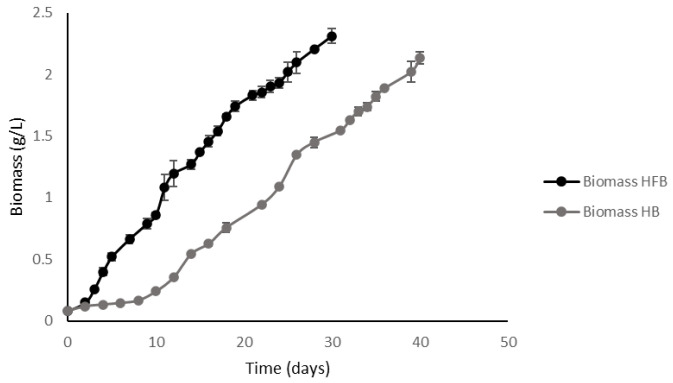
Biomass accumulation in fed-batch and batch culture systems. The figure shows the biomass accumulation kinetics (g/L) as a function of time for cultures in batch mode and fed-batch mode under the same culture medium conditions. Cultures were stopped 4 days after the nitrogen depletion. HB: high-density batch. HFB: high-density fed-batch.

**Figure 6 plants-11-03229-f006:**
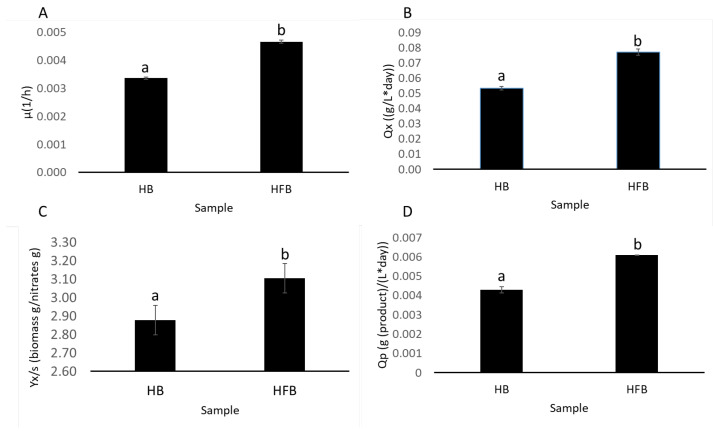
Comparative analysis of different kinetics and production parameters for *M. salina* cultures in different modes of cultivation and amount of nutrients. In this section of the results, all parameters were calculated 4 days after the nitrogen depletion in the growing medium and compared the results by ANOVA statistical analysis (*p* < 0.05), “**a**” and “**b**” denotes significant differences. (**A**) Specific maximum growth rate. (**B**) Biomass productivity. (**C**) Substrate yield. (**D**) Volumetric productivity of β-glucans. HB: high-density batch. HFB: high-density fed-batch.

**Table 1 plants-11-03229-t001:** Parameters used to calculate the experimental time in fed-batch culture.

Parameter	Magnitude
Yx/s	6.2000
si [g/L]	0.7040
s [g/L]	0.0400
b	0.2429
μ [h^−1^]	0.0052
Vo [L]	0.5000
V [L]	1.5000
Xo [g/L]	0.3200
t [h]	631.8820

**Table 2 plants-11-03229-t002:** Biochemical parameters 4 days after nitrogen depletion in culture medium.

Compound	High-Density Batch (% Dry Weight)	High-Density Fed-Batch (% Dry Weight)
Lipids	43.72 ± 1.33	40.20 ± 2.20
Carbohydrates	17.23 ± 0.15	16.23 ± 1.78
Proteins	10.23 ± 0.68	12.53 ± 0.92
β-glucans	8.04 ± 0.14	7.71 ± 0.43

## Data Availability

The data presented in this study are available in Appendix A.

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
