# Peer review of "A Modeled High-Density Fed-Batch Culture Improves Biomass Growth and β-Glucans Accumulation in Microchloropsis salina"

_plants, 2022, doi:10.3390/plants11233229_

Round 1

Reviewer 1 Report

The manuscript has focused on the cultivation conditions and biomass characteristics. The experimental design has covered sufficient aspects of the cultivation process. The results were reasonable. I would like to recommend the publication of this study if the following improvements could be made.

1)    It is difficult find a novelty in this study. How is it different from previous studies, and why do author think this study is necessary?

2)    All the figures are very difficult to be understand, please insert a legend.

3)    Please insert Chapter 4. Material and methods, after Introduction. – - for a good understanding of the work.

4)    Please cite in the text (especially in the Introduction part) – current references ( 2022, 2021, 2020).

Reviewer 2 Report

The manuscript is poorly written. It will be very difficult to understand the results in its current version. Also, references are not updated. The authors need to be included updated references.

The authors used many non-stop long sentences throughout the manuscript. For example, “In the case of euglenoids, that are capable of accumulating high quantities of paramilon, a type of linear β-glucan that is stored in form of granulocytes [36] or in diatoms such as Skeletonema costatum that is also capable of accumulating β-glucans as an energy reservoir in stationary stages of development [37], it has been suggested a mechanism in which the microorganisms diminishes its metabolisms and concentrates in accumulating reserve metabolites to maintain cell viability”.

“In this research it was possible to characterize the growth of Microchloropsis salina in low and high density batch cultures, which allows to establish parameters for the modeling of a two-stages fed batch culture, which together with the modification of environmental parameters have proved to be an effective alternative to increase the productivity of metabolites of interest like lipids, carbohydrates and β-glucan, as well as obtaining high yields of biomass, shortening the periods of experimentation without having to modify genetics to meet these objectives”.

There are also English errors throughout the manuscript and they should be resolved before resubmission. 

Round 2

Reviewer 2 Report

The manuscript is much improved after revision. However, writing in the current version is still poor. Still, there are lots of long sentences throughout the paper and those should be fixed before publication. 
